# Enhanced Terahertz Sensing via On-Chip Integration of Diffractive Optics with InGaAs Bow-Tie Detectors

**DOI:** 10.3390/s25010229

**Published:** 2025-01-03

**Authors:** Karolis Redeckas, Vytautas Jakštas, Matas Bernatonis, Vincas Tamošiūnas, Gintaras Valušis, Linas Minkevičius

**Affiliations:** 1Department of Optoelectronics, Center for Physical Sciences and Technology (FTMC), Saulėtekio Ave. 3, LT-10257 Vilnius, Lithuania; 2Institute of Photonics and Nanotechnology, Vilnius University, Saulėtekio Ave. 3, LT-10257 Vilnius, Lithuania

**Keywords:** terahertz sensing, bow-tie detectors, on-chip integration, Fresnel zone plates, sensitivity enhancement

## Abstract

The practical implementation of terahertz (THz) imaging and spectroscopic systems in real operational conditions requires them to be of a compact size, to have enhanced functionality, and to be user-friendly. This work demonstrates the single-sided integration of Fresnel-zone-plate-based optical elements with InGaAs bow-tie diodes directly on a semiconductor chip. Numerical simulations were conducted to optimize the Fresnel zone plate’s focal length and the InP substrate’s thickness to achieve constructive interference at 600 GHz, room-temperature operation and achieve a sensitivity more than an order of magnitude higher—up to 24.5 V/W—than that of a standalone bow-tie detector. Investigations revealed the strong angular dependence of the incident radiation on the Fresnel zone plate-integrated bow-tie diode’s response. These findings pave a promising avenue for the further development of single-sided integration of flat optics with THz detectors, enabling improved sensitivity, simplified manufacturing processes, and reduced costs for THz detection systems in a more compact design scheme.

## 1. Introduction

Terahertz (THz) frequencies, situated between the microwave and infrared ranges in the electromagnetic spectrum, present fascinating research opportunities and exhibit a broadband scale of versatile applications. From a device physics perspective, they bridge the classical carrier transport employed in millimeter waves and the operation of photonic devices, defined by quantum mechanical rules. This particular crossover is stimulating the development of novel devices via the confluence of different physical principles to enable their operation. From an application point of view, THz radiation manifests a remarkable ability to penetrate non-conductive materials opaque to visible light, hence making THz imaging and spectroscopy powerful non-invasive and non-destructive investigation tools for diverse fields, such as materials science, biomedical diagnosis, and security screening [1,2]. Furthermore, for objects with relatively low absorption, coherent measurement techniques such as heterodyne [3,4] or homodyne [5,6] detection can facilitate the identification of the composition of materials or assessments of their quality through observing the induced phase variations.

The implementation of THz imaging and spectroscopic systems in real operational conditions requires optimization and a compact size such systems should feature enhanced functionality and sensitivity, reduced power consumption, and convenience in terms of their use [2]. Increased detection capabilities in THz imaging systems can be achieved by exploring new approaches to detection improvements—for instance, demonstrating quantum cascade lasers operating as resonantly amplified THz detectors with a wide RF bandwidth [7] or employing new materials, such as HgTe, in bow-tie antenna-integrated THz sensors [8]. An interesting route may also be exposed via the use of MgB_2_ in hot electron bolometers [9] or advanced micromechanical bolometers containing a Si_3_N_4_ trampoline with a 35 nm thick Cr-Au coating [10], which enables a high-speed response, low noise, and highly stable operation in ambient conditions, i.e., at room temperature. Notable developments also include room-temperature, zero-bias, Schottky-diode-based THz detectors [11] for frequencies up to 5.56 THz; the demonstration of biquad antenna coupling to achieve a low noise-equivalent power (NEP) in CMOS detectors for broadband sensing around 1 THz [12]; and modeling the power coupling of bow-tie antennas to the FET channel, resulting in a greater than double performance improvement in high-sensitivity AlGaN/GaN HEMT detectors [13]. An exciting way to achieve enhanced functionality is to enrich the operation of THz imaging systems by implementing machine learning methods in object recognition [14] or by applying a flexible illumination scheme based on fast beam steering, thus enabling an improved signal-to-noise ratio, homogeneous and adjustable illumination, and an optimized distribution of the available power [15]. The most effective way to achieve ease of use and reduced power consumption simultaneously is to integrate the active and passive elements of the imaging systems directly onto a semiconductor chip. Examples of innovative solutions in this direction include on-chip integration of a plasmonic FET-based source and a nano-patch antenna for efficient THz emissions [16] or the implementation of a THz lens antenna for a photonic integrated circuit-based THz system [17], thus marking a stimulating shift in the intense evolution of integrated THz photonic devices [18].

Our work advances this field by proposing the single-sided integration of Fresnel-zone-plate-based optical elements and InGaAs bow-tie sensors directly onto a semiconductor chip. The bow-tie diodes, exhibiting broadband—up to 2.5 THz—and room-temperature detection [19] of THz radiation and a fast (less than 7 ns) response time [20], together with the Fresnel zone plates revealing themselves as effective focusing elements for THz radiation [21], fully complement each other in a flat solution for THz light collection and detection. In contrast to previous work [22], which provided both increased sensitivity and a new pathway to simplified and robust THz detection systems through the rigid integration of the detector with secondary optics on different sides of the substrate, the presented solution provides greater flexibility in the electric circuit design and opens a more convenient pathway to readout and processing of the detected signals and flip-chip integration. Furthermore, in contrast to the direct laser writing employed in [22], the zone plates were processed using photolithography and wet patterning [23]. This not only eliminated the alignment issues for the zone plates and detectors but also allowed for a fourfold increase in the radiation collection area compared to that with a double-sided processing scheme for the same thickness of substrate. Moreover, it opens new avenues for the low-cost mass production of such detectors for room-temperature THz imaging systems and related applications.

## 2. Materials and Methods

### 2.1. Numerical Simulations

CST Studio Suite with the program package “Microwave Studio”, which utilizes the finite integration technique (FIT), was employed to simulate the THz light propagation and the spatial profile of the incident THz radiation of the binary integrated zone plates (ZPs) surrounding the bow-tie (BT) detector. The simulation parameters, such as the plane-wave frequency *f* = 600 GHz, the electric field strength E = 1 V/m, the linear polarization along the bow-tie structure (the *x*-axis), and perfectly matched layer boundary conditions, which represented open space to minimize the reflection at the edges of the simulated volume, were fixed during all of the simulations. The resolution throughout the structure, defined by the mesh density, was set to 16 lines per wavelength. When modeling the BT structure, the local mesh option was used, and a mesh density of 1000 lines per wavelength was specified for the active region of the detector.

A four-step modeling approach was applied to fully comprehend the electromagnetic coupling and underlying physics of the sensors’ operation. This methodology enabled us to break down the optimization processes into individual steps and thoroughly examine the specific role of each integrated part.

In the first modeling step, the simulations take into consideration the concentric gold rings on the InP substrate defined by the formula for Fresnel zone plates [24]. The substrate has another flat opposite side fully covered by a 20 μm golden layer to reflect the THz radiation, as shown in the left-hand panel of Figure 1. The optimization process involved varying the thickness of the InP substrate (*d*) between 326 μm and 521 μm, representing commercially available substrates, with corresponding adjustments to the zone plate’s focal distance (*F*) according to the relation (*F* = 2*d* ± *const.*). An additional value, *const.*, was incorporated to accurately account for the constructive interference with the incoming radiation to the center of the zone plate at the surface and to compensate for the reflection-induced phase shift. The focusing optical element consisted of *N* = 12 sub-zones—6 golden rings and 6 rings transparent to THz light. The design followed previous findings, allowing us to reach the highest efficiencies with the zone plates [25].

The second modeling step was focused on evaluating the sensing capabilities of the BT diode as a single device, specifically examining how its asymmetric contact geometry concentrates the electric field into the structure’s neck. The diode design consisted of a 500 nm thickness InGaAs BT structure (permittivity: *ϵ_InGaAs_* = 11.6) on the InP (permittivity: *ϵ_InP_* = 9.52) substrate, with the backside covered by a thin gold layer. The geometry of the individual diode for 600 GHz detection was chosen according to [26]—*b* = 5 μm, *L* = 100 μm, *W* = 25 μm, *I* = 40 μm, and *a* = 10 μm (see the right panel of Figure 1)—and the InGaAs was covered with 200 nm gold, which was treated as lossy metal, with *σ_Au_* = 4.561 × 10^7^ S/m cap.

The third modeling step incorporated the previously defined BT structure with straight rectangle golden stripes that connected the diode and a single split golden sub-zone around it. These components served as a contact pad, enabling us to observe the full zone plate’s utility and resolve the electrical contact’s possible influence.

The fourth modeling step examined the BT detector with contact pads and all 12 sub-zones of the Fresnel ZP. This comprehensive model permitted an illustration of the enhancement in the electric field in the active zone of the BT, which was integrated with a flat focusing optical element.

### 2.2. Sample Growth and Processing

The BT detector with an integrated ZP is shown in Figure 2. The unintentionally doped 530 nm thick lattice-matched In _0.53_Ga_0.47_ As layer was grown on a 335 μm thick Fe-compensated semi-insulating InP substrate of a 2” diameter (Wafer Technology Ltd., Bucks, UK) in the Veeco GENxplor^®^ molecular beam epitaxy system (Veeco Instruments Inc, St. Paul, MN, USA). The average carrier density and mobility were determined using Hall effect measurements, with an induction of the magnetic field of 0.944 T. The 5.46 × 10^14^ cm^−3^ and 7710 cm^2^/(Vs) values were obtained at room temperature.

The samples were fabricated using two-step UV lithography, wet mesa etching, and a lift-off process with sputtered thin metal films. The SUSS MA/BA6 Gen4 mask aligner (SÜSS MicroTec, Garching, Germany) and the AZ5214E image-reversal photoresist (Merck Performance Materials GmbH, Wiesbaden, Germany) were used to form a pattern for the mesa etching. A phosphoric acid peroxide mixture was prepared by adding hydrogen peroxide (H_2_O_2_, 30%, Sigma-Aldrich, St. Louis, MO, USA) to the phosphoric acid (H_3_PO_4_, 85%, Sigma-Aldrich) and diluting it with deionized water. The solution with a constant 1:1:10 volumetric ratio had an etching rate of ≈6 nm/s. A DektakXT stylus profilometer was used to measure the thickness of the patterned photoresist before and after etching to ensure that all of the InGaAs layers were etched onto the substrate. A similar second step of photolithography was employed to form a pattern for metallization. A metal stack comprising 20 nm of Ti and 180 nm of Au was evaporated using a TFDS-870 (VST Service Ltd., Eastleigh, UK) e-beam sputtering system, and the structures were processed through a lift-off procedure using dimethyl sulfoxide (DMSO). A Unitemp RTP-100-HV rapid thermal process oven was used to anneal the metal contacts for 10 s at a 400 °C temperature in ambient nitrogen. Profile measurements were used to verify the thickness of the mesa and the metallization, as shown in Figure 2. Afterward, the fabricated structures were transferred onto a specially designed PCB board for convenient detector implementation and imaging. The antenna was connected to the PCB by adhering it to 25 μm gold wires using ultrasonic wire bonding (TPT HB10, Munich, Germany).

### 2.3. Electrical Characterization

The manufactured BT diode detectors underwent initial electrical testing by measuring the direct current (DC)−voltage (I–V) characteristics to assess their basic electrical properties and ensure their proper functionality. These measurements were conducted using a Cascade Microtech EPS150TRIAX (Beaverton, OR, USA) probe station and a Keithley Instruments (Solon, OH, USA) 2400 source measurement unit. Figure 3 depicts the I–V characteristics of the measured BT detector with a zone plate (BT + ZP) and that without it. The linear fit of the I–V curves near zero bias provided resistances of 4400 Ω and 5100 Ω for the BT and BT + ZP samples. The I–V curves exhibited asymmetry, which was quantified using coefficients calculated as *A* = (*I*_F_ − *I*_B_)/(*I*_F_ + *I*_B_), where *I*_F_ and *I*_B_ were the forward and reverse current. The current and its asymmetry coefficient compared to the applied voltage are shown in Figure 3. It is important to note that the BT diode operates according to a passive detection scheme, i.e., no bias voltage is needed. Consequently, the backward (leakage) current *I*_B_ cannot have a substantial effect on its operation. The asymmetry observed in the BT diode justifies its operational principle, which relies on non-uniform electron heating. This asymmetry stems from the diode’s particular structure, with one part of the BT diode composed of metal and the other of a bare semiconductor of an asymmetrical shape [19,26]. Some of the deviations in the I–V curves and the asymmetry coefficient’s dependence on the applied voltage might be attributed to the uneven etching of the mesa since the resistance of the BT diode is strongly dependent on its geometrical parameters, particularly the width of the neck.

### 2.4. THz Characterization Setup

A commercially available VDI AMC364 electronic frequency multiplier chain served as the THz radiation source of a 600 GHz frequency, delivering power up to 905 μW. The radiation generated by the Quonset Microwave QM2010-24-27 synthesizer was used as a source for frequency multiplication. The intensity of the THz radiation was electrically modulated at a 1 kHz frequency using an Agilent 33519B waveform generator (Agilent, Santa Clara, CA, USA). The divergent beam from the THz source was first collimated and then focused onto the fabricated structure using two off-axis parabolic mirrors of a focal length *F* = 10 cm and a diameter *D* = 5 cm. The signal amplitudes of the BT detectors were recorded using an SR7265 lock-in amplifier (Signal Recovery, AMETEK, Inc., Oak Ridge, TN, USA). The detector positioning was achieved through a computer-controlled 3-axis positioning system. Two Thorlabs DDS300 motorized stages were used for *xz* positioning with an absolute on-axis accuracy of 7.5 μm. At the same time, the positions along the third *y*-axis were adjusted using a Thorlabs LTS300C motorized stage. In addition, a mechanical rotational stage was used in some of the experiments to adjust the rotation angle in 2.5° increments. Section 3.2 provides more details on the experiments performed.

## 3. Results

### 3.1. Simulation Results

As previously mentioned in Section 2.1, the simulations started with the Fresnel zone plate, which was incorporated into the InP wafer, with the gold-covered backside of the substrate acting as a reflecting mirror, as shown in the left-hand picture in Figure 4a. The highest efficiency, which manifested as an electric field strength increased 13.4 times in focus, was achieved with the InP substrate’s thickness *d* = 355 μm and the Fresnel zone plate’s focal distance *F* = 645 μm (the central picture in Figure 4a). This met the requirements of multiple half-wavelengths, resulting in constructive interference of the incident and the reflected beams being at the center of the zone plate. According to the simulation results (the right picture in Figure 4a), THz wave interference occurs throughout the InP wafer, with the largest constructive maxima of 13.9 V/m located close to the surface. Therefore, such a design allowed us to achieve the highest electric field strength on the surface.

The simulation of the second step included the already defined (see Section 2.1) BT detector and the thickness of the InP substrate with a gold-covered backside *d* = 335 μm. The conditions remained consistent with the previous case, as shown in the left-hand picture in Figure 4b. The simulations were conducted on a substrate with the same width *w* = 4000 μm, identical to that in the first model. The simulation results on the electric field’s distribution in the structure are depicted in Figure 4b. It is seen that the BT diode, due to its geometrical shape, enables the electric field to be concentrated in the neck region. This leads to an 8.1 times higher local electric field strength than that of the incident electric field and an 1.65 times lower strength than that achieved by reflection (Figure 4a).

In the third modeling step, the model was expanded to include the previously simulated BT detector with integrated gold contact pads (the left picture in Figure 5a), which were needed to record the electrical signal. According to the simulation results (the right picture in Figure 5a), this design allowed us to achieve a 55.3 V/m electric field strength at the narrowest point in the active InGaAs layer, which was 6.8 times higher compared to that with the single BT diode on the substrate (Figure 4a). The electric field is highly localized at the edge of the metal, around the narrowest point of the BT sensor, as seen in the *xz* plane (bottom right in Figure 5a).

The fourth step examined a complete structure combining a full Fresnel ZP structure (from the first modeling step) and the BT diode with contact pads (from the third model) (left picture in Figure 5b). The results obtained demonstrated that the high electric field can reach values of 250 V/m in the narrowest part of the BT diode in the InGaAs layer (top-right in Figure 5b), resulting in electron heating in the neck of the device. This electric field is ≈31 times higher compared to that with the single BT diode (top-right in Figure 4b) and 4.5 times higher compared to that with the BT diode with a single integrated zone and contact pads (top right in Figure 5a), showcasing the significant potential of the enhanced detector’s sensing capabilities.

### 3.2. THz Characterization

To reveal the influence of the integrated optics on the detection performance, experimental measurements were made in two detector configurations: with the BT detector with the integrated Fresnel zone plate and with the standalone BT diode, acting according to its asymmetrical shape, and contact pads. These configurations corresponded to the fourth and third simulation steps, respectively. The initial measurements involved raster scanning of the focused beam in the *xy* and *xz* planes, as depicted in the experimental setup (Figure 6a). The focused THz beam scanning results depicted in Figure 6b highlight the significant 18.9-fold increase in the maximal detected signal value compared to that in the standalone BT diode case. An improvement of this order aligns well with the simulation results, as optical power maintains a square proportional relationship to the electric field strength [27].

The detectors’ responsivities were evaluated by considering the size of the effective detector area. In more detail, the BT diode with integrated optics, which has an effective detector area of *S*_Zones_ = 6.9 × 10 ^−6^ m^2^, reaches a responsivity *R*_Zones_ = 24.5 V/W. This value indicates a 10.6-fold improvement compared to the BT diode without a Fresnel zone plate, whose effective area is *S*_No_zones_ = 7.4 × 10^−7^ m^2^ and whose responsivity amounts to *R*_No_zones_ = 2.3 V/W. This enhancement in sensitivity can be attributed to the improved coupling efficiency of the THz radiation with the BT diode’s active region without altering the noise level. The capability of the detectors to detect low-power THz radiation was evaluated using estimates of the noise-equivalent power (*NEP*). Indeed, the BT with integrated optics achieved an *NEP*_Zones_ = 3.76 × 10^−10^ W/$\sqrt{\mathrm{Hz}}$, which was an order of magnitude lower than the *NEP*_No_zones_ = 3.78 × 10^−9^ W/$\sqrt{\mathrm{Hz}}$ achieved by the single BT diode. Such results were obtained through careful consideration of both the detector size and the BT diode’s resistance. The increase in responsivity indicates important improvements achieved through the integration of the Fresnel ZP.

To confirm and illustrate the effect of the detection improvement caused by the integrated Fresnel zone plate, the dependence of the detected signal on its angle of incidence to the BT diode’s plane was experimentally investigated. For this study, an additional rotational axis was employed, and the incident radiation to the optical axis was adjusted in 2.5° increments. The dependency of the maximum signal value on the angle scanning in the *xy* plane for diodes with and without integrated optics is displayed in Figure 7. The results showcase a noticeable decrease in the signal with the rotational angle for the detector with integrated optics, which can be described according to a slope in the linear region Θ_Zones_ = −7.23 μV/deg. This dependence occurs due to the increasing phase mismatch of the plane wave incident on the surface, resulting in reduced focusing efficiency of the Fresnel ZP and the shifted position of the focal point. In contrast, this dependency of the signal on the angle is not pronounced Θ_No_zones_ = −1.89 μV/ for the BT diode with contact pads, which maintains a similar effective antenna area with a changing incident angle of radiation and is weakly affected by a single zone around it. Additionally, to confirm this tendency of the signal being dependent on the angle, a numerical simulation of the BT diode with the integrated zone plate was conducted. The results presented in Figure 7 show the similar tendency of the electric field on changing the angle. This dependence clearly indicates that the enhancement in the detection capabilities requires precise alignment.

## 4. Conclusions

This study exhibits the enhancements in THz detection when using single-sided integration of Fresnel zone plate optics with InGaAs bow-tie diodes, achieving a substantial increase in sensitivity. The design of the integrated optics allowed them to achieve a 31 times higher electric field, resulting in a 10.9 times enhanced sensitivity, reaching 24.5 V/W, and a noise-equivalent power lower by an order of magnitude compared to the standalone bow-tie diode with contact pads. The dependence of sensitivity on the angle of incident illumination revealed a pronounced response in the integrated system, confirming the role of the Fresnel zone plate optics in enhancing the detected signal. The investigation’s results expose that single-sided integration of flat optics into the detector not only enables enhanced sensitivity and functionality but also simplifies the fabrication technology and reduces the costs of THz detection systems while maintaining a compact design and a user-friendly operation scheme.

To finalize and extrapolate our findings, the technological approach to the integration of Fresnel zone plates and bow-tie diodes on the same side of the substrate demonstrated not only offers more freedom in designing further integrations for the sensors but also extends their potential applications. In particular, thanks to bow-tie diodes’ ability to operate in the heterodyne mode [4], their broadband range [19], and their fast response regime [20], they allow end-users to customize their end function and provide further integration/miniaturization pathways. Therefore, the proposed approach shows promise as a significant contributor to metasurface integration schemes [28,29], thus making THz medical imaging, for instance, more accesible and convenient [30].

## Figures and Tables

**Figure 1 sensors-25-00229-f001:**
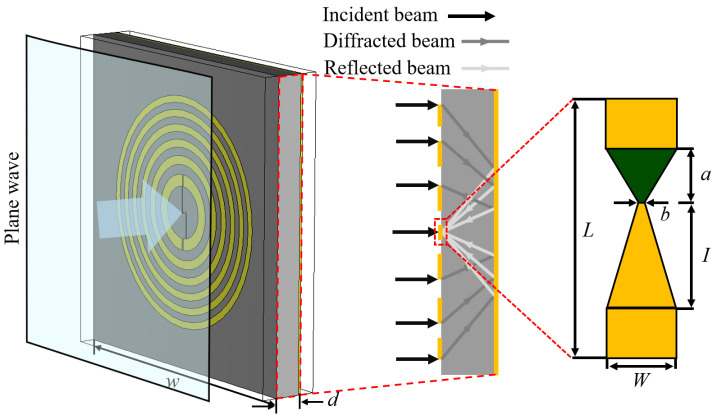
Sketch depicting the model of a bow-tie diode (BT) with an integrated Fresnel zone plate employed for the simulations. Left panel—schematic of the sensor with integrated Fresnel zone plate with incident plane THz wave. The rectangular parallelepiped around the model represents the modeling volume used in the simulations. Geometry of the model: *w* = 4000 μm, width; *d*—thickness of the substrate. The middle panel illustrates the THz wave propagation and focusing. The right panel shows the enlarged BT diode and its geometrical parameters: *b* = 5 μm, *L* = 100 μm, *W* = 25 μm, *I* = 40 μm, and *a* = 10 μm.

**Figure 2 sensors-25-00229-f002:**
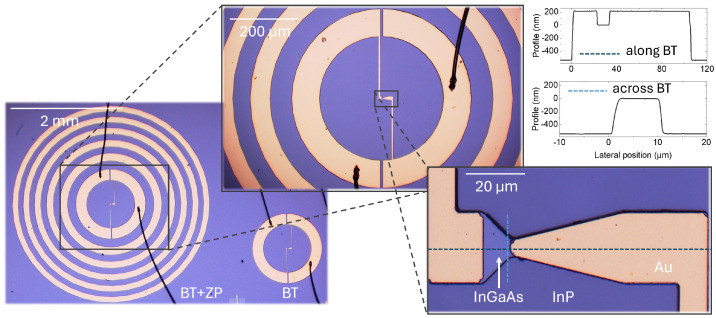
Micrographs of the fabricated bow-tie detector with an integrated zone plate, designed for a 600 GHz frequency (BT + ZP), and the reference BT detector only with the contact pads. The left panel shows the entire structure, highlighting both the BT detector and the ZP elements. The center panel provides a zoomed view of the central part with a 200 μm scale bar and zone plate rings and contact pads. An enlarged image of the InGaAs BT detector’s active part and gold electrodes is shown in the right panel, with a 20 μm scale bar. Surface topographic cross-sections of the fabricated BT diode, depicted in the top-right corner, were measured along the length of the BT and across the BT using a DektakXT profilometer. Profile measurement lines are marked with dashed lines in the right panel. The measured profiles are shown in the insets.

**Figure 3 sensors-25-00229-f003:**
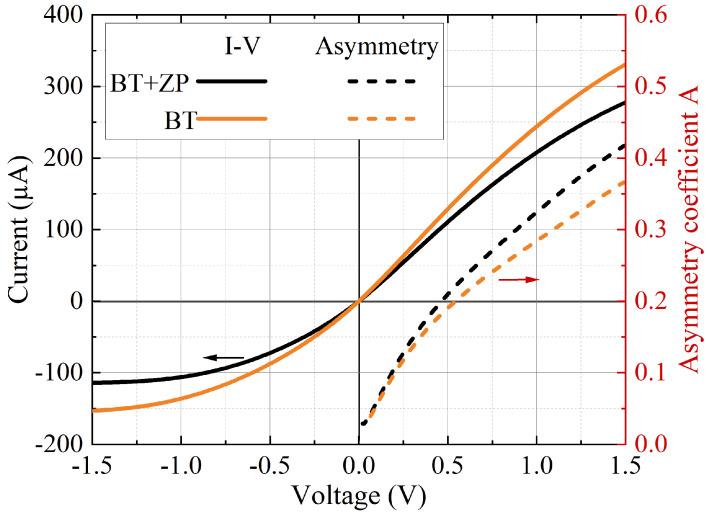
I–V characteristics of the BT detector with an integrated zone plate (BT + ZP; solid black line) and that without one (BT; solid orange line). Note the asymmetry of the I–V characteristics expressed via the asymmetry coefficient ($A=\left(I_{F}-I_{B}\right)/\left(I_{F}+I_{B}\right)$), which is dependent on the applied voltage (dashed lines).

**Figure 4 sensors-25-00229-f004:**
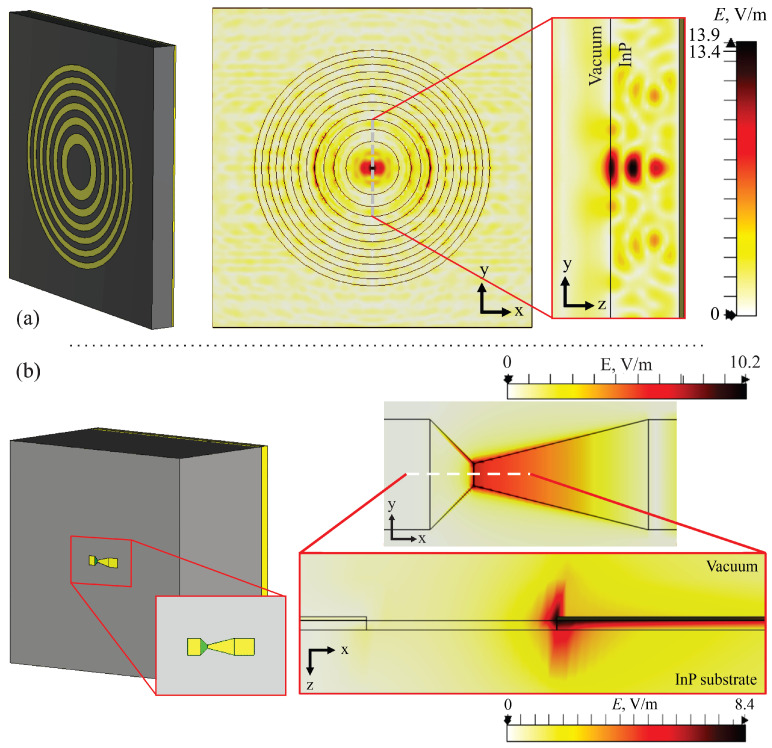
A general view of the structure and the simulation results. Panel (**a**): Left picture—a single Fresnel zone plate. Note that the coverage of the backside of the substrate in gold is denoted in yellow. The central picture displays the distribution of the electric field strength in the *xy* plane. Right picture—enlarged part of the electric field’s distribution close to the sensors in the *yx* plane. Note the maximum interference in the vicinity of the sensor. Panel (**b**): a single BT diode (left picture) and the distribution of the electric field in the vicinity of the neck in the *xy* and *zx* planes (center top and bottom pictures, respectively). Simulations were performed on the surface of the substrate (*z* = 0 μm) for the single Fresnel zone plate and on the InP substrate in the case of the single BT diode (*z* = −0.5 μm). The dotted line represents the center of the simulated structure. The thickness of the InP substrate *d* = 335 μm; the focal length of the Fresnel zone plate *F* = 645 μm. Note the substantial increase in the electric field and its non-uniformity in the neck of the structure due to the asymmetrical design.

**Figure 5 sensors-25-00229-f005:**
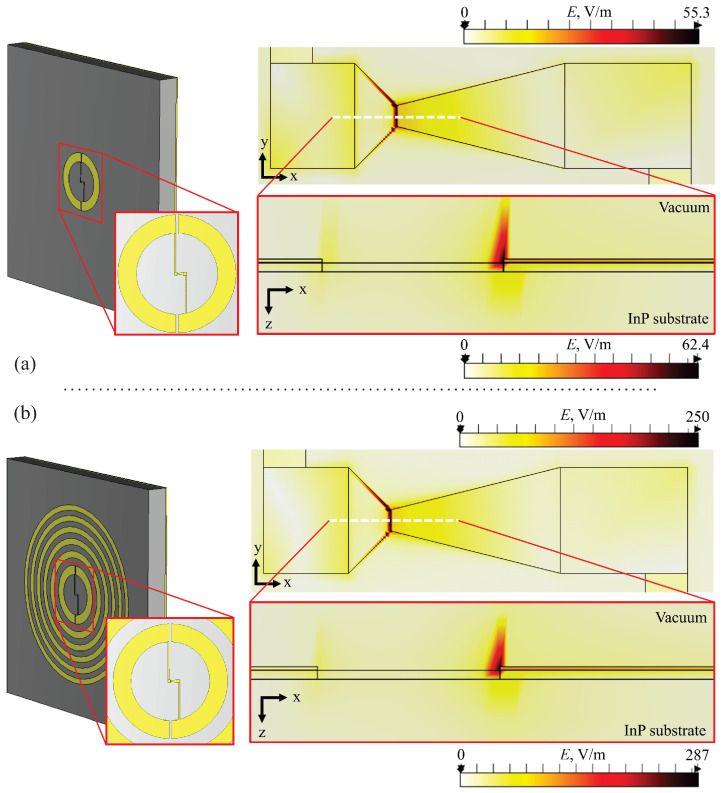
General view of the structure and simulation results. Panel (**a**): Left picture—schematics of the BT diode and split zone contact pads. Right picture—distribution of the electric field strength in the *xy* plane (top part) and an enlarged cross-section in the *xz* plane (bottom part) in the active area of the BT diode placed on the InP substrate (*z* = −0.5 μm). The dotted lines represent the center of the simulated structure. Panel (**b**): Left picture—schematics of the BT diode, contact pads, and the Fresnel zone. Right picture—distribution of the electric field strength in the *xy* plane (top part) and an enlarged cross-section in the *xz* plane (bottom part) in the active area of the BT diode placed on the InP substrate (*z* = −0.5 μm). Note the strong increase in the electric field in the neck of the structure. Its triangle-like shape illustrates the non-uniformity of the electric field. The thickness of the InP substrate is *d* = 335 μm.

**Figure 6 sensors-25-00229-f006:**
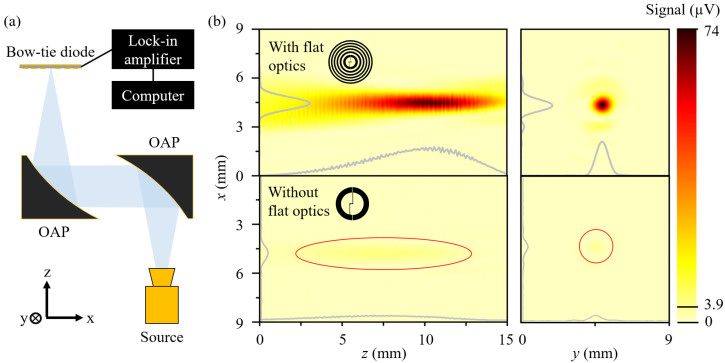
Panel (**a**) represents the experimental setup and schematic connection used for focused beam imaging. The OAP denotes the off-axis parabolic mirrors. Panel (**b**) depicts the distribution of the intensity of the focused beam, registered when employing the on-chip integrated BT diode and the Fresnel ZP (top part) and the BT diode with contact pads only (bottom part). Left side—measurements along the optical axis in (*xz*), where the focal depth of the off-axis parabolic mirror with a 10 cm reflected focal length is scanned. Right side—measurements in the focal plane (*xy*). Red lines serve as a guide for the eyes to indicate focusing areas of the 3.9 μV signal, which is not strongly pronounced in the case with the standalone BT diode.

**Figure 7 sensors-25-00229-f007:**
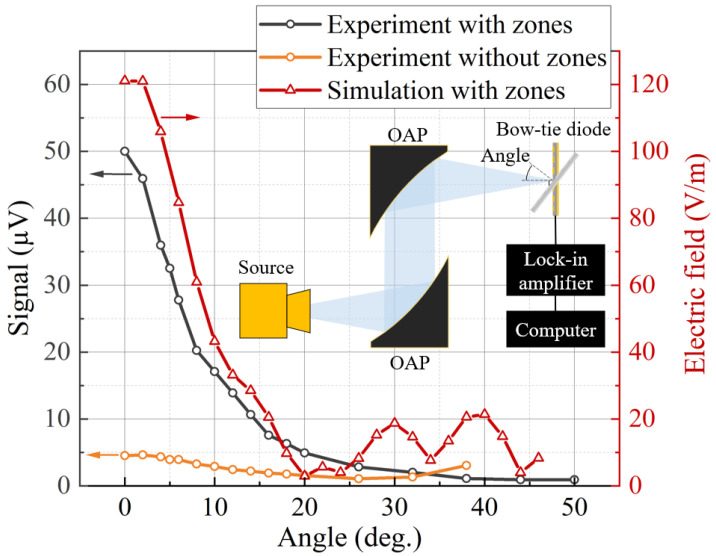
Dependence of the detected signal of the structure on the incident angle. The gray line represents the BT diode with the integrated Fresnel ZP and the orange line that without it. The simulation results are given for comparison. The inset presents the setup and schematic connection used to measure the dependence of the signal on the incident angle of the THz radiation. The OAP denotes the off-axis parabolic mirrors.

## Data Availability

The data that support the findings of this study are available from the corresponding author upon reasonable request.

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
