# Peer review of "Enhanced Terahertz Sensing via On-Chip Integration of Diffractive Optics with InGaAs Bow-Tie Detectors"

_sensors, 2025, doi:10.3390/s25010229_

Round 1

Reviewer 1 Report

Comments and Suggestions for Authors

The work is devoted to improving the characteristics of terahertz (THz) detectors through the integration of Fresnel plates. The research is relevant in the context of the rapid development of THz radiation and detection technologies, which find applications in areas such as biomedical diagnostics, material quality control, and security systems. The authors demonstrate significant improvements in sensitivity and reduced noise equivalent power due to the integration of optical elements.

However, before recommending the article for publication, substantial revisions should be made to the text:

  1. The justification for using an In0.53Ga0.47As film with a thickness of 530 nm and an InP substrate with a thickness of 335 µm is unclear in the article. It would be helpful to explain why these specific parameters were chosen and to provide references to relevant studies or experimental rationale.
  2. Has the photoconductive layer been characterized beyond the measurement of carrier mobility, for example, by determining the lifetime of photogenerated carriers? This is important for understanding the film's quality and its contribution to the overall device performance.
  3. If Figure 3 indeed reflects the values of dark current, values in the hundreds of µA appear high. This may indicate significant leakage currents, which require discussion of their impact on the device's performance.
  4. The statement that an increase in the electric field in the focus by 13.4 times was achieved with a thickness of d = 335 µm and a focal length of F = 645 µm is unclear. It is necessary to specify what is meant by d = 335 µm.
  5. The simulation results presented in Figure 4(a) look unusual due to the presence of white/light-yellow "streaks." Does this have a physical meaning, or is it related to the computational mesh?
  6. The periodic red-burgundy bands in the image raise questions about their origin. Why do they propagate similarly in both the vacuum and the InP layer? This requires clarification.
  7. The representation of the electric field as a "triangle" in the vacuum raises questions. Is this a real result or an artifact of the calculations? If it is a physically accurate result, the nature of such a field distribution should be explained.

Author Response

We are thankful for Your remarks and questions.

Please find our answers provided in the attached PDF file .

Reviewer 2 Report

Comments and Suggestions for Authors

Karolis and co-authors demonstrate InGaAs detectors for THz sensing application. The device is realized by integrating the bow-tie antenna with Fresel zone plate on a single chip. By numerically optimizing the geometry of the focal length of the Fresel zone plate, they successfully achieve more than one order of magnitude higher sensitivity compared to single bow-tie design. The following experiments were conducted to support such an achievement. The paper can be accepted by Sensors after taking care of some concerns:

1.      When conducting numerical simulations in CST microwave studio, the material parameters, including the refractive index, permittivity, or conductivity of the InGaAs, the gold (Drude model?), and the InP substrate, are not clear. The authors are strongly recommended to add corresponding details.

2.      Talking about optimization, I didn’t find any optimization strategy or algorithm in the manuscript. Will the authors explain how to conduct optimization? Moreover, the geometric paraments of the zone plate is not provided.

3.      When explaining the THz characterization setup (section 2.4), the schematic connection is recommended to be added for better understanding how to conduct the measurement.

Basically, the authors have presented a THz detector. But practical sensing applications, like near-field coupling to biomolecular or chemical analyte, e.g., PhotoniX volume 5, Article number: 10 (2024), PhotoniX volume 4, Article number: 28 (2023), and PhotoniX volume 2, Article number: 12 (2021), or imaging THz sensitive materials e.g., PhotoniX volume 5, Article number: 2 (2024) and PhotoniX volume 1, Article number: 12 (2020) are not practically conducted yet. The authors are recommended to discuss if such device can be helpful for THz near-/far-field bio-/chemical sensors. I think it could be quite interesting to employ their presented active device for these applications.

Comments on the Quality of English Language

N/A

Author Response

(The authors gave the same response as above.)

Round 2

Reviewer 1 Report

Comments and Suggestions for Authors

My comments and suggestions were taken into account during the process of correcting the article. I believe that the article in this form can be accepted for publication.